# Making Digital Objects FAIR in High Energy Physics: An Implementation for `Universal FeynRules Output (UFO)` Models

Mark S. Neubauer[1], Avik Roy[1*] and Zijun Wang[1]

**1** University of Illinois at Urbana-Champaign
* avroy@illinois.edu

March 17, 2023

## Abstract

Research in the data-intensive discipline of high energy physics (HEP) often relies on domain-specific digital contents. Reproducibility of research relies on proper preservation of these digital objects. This paper reflects on the interpretation of principles of Findability, Accessibility, Interoperability, and Reusability (FAIR) in such context and demonstrates its implementation by describing the development of an end-to-end support infrastructure for preserving and accessing `Universal FeynRules Output (UFO)` models guided by the FAIR principles. UFO models are custom-made python libraries used by the HEP community for Monte Carlo simulation of collider physics events. Our framework provides simple but robust tools to preserve and access the UFO models and corresponding metadata in accordance with the FAIR principles.

# 1  Introduction

Much of ongoing research in the discipline of high energy physics (HEP) require using high-end computational resources. Examples include analyzing petabyte scale data collected at the Large Hadron Collider (LHC) [1] or obtaining precise theoretical predictions that may require taking into account matrix element calculation from thousands of Feynman diagrams [2]. Being such a computationally intensive research discipline, HEP research uses different kinds of digital objects (DOs). These DOs used in HEP include large datasets obtained from detector operation or Monte Carlo (MC) simulations, data analysis software like ROOT [3], MC simulation software like Pythia [4], Sherpa [5], Herwig [6], and MadGraph [7], dedicated libraries like LHAPDF [8] for calculating the parton distribution functions, as well as numerous privately developed codes and software packages, documentations, tutorials and notebooks. While collaborations facilitate the preservation of their data and common software frameworks for usage both within and outside of the collaboration[1], preservation of digital resources independently developed by smaller research groups or individuals is equally important to be able to reproduce the results from HEP research. Hence, a significant amount of community-wide effort and resources has been put forward in order to preserve these DOs. For instance, the Durham High-Energy Physics Database (HEPData) [11] is an open-access repository established for preserving and sharing scattering data from HEP experiment, containing digitized details of plots and tables from thousands of physics analysis publications. Similar efforts have been put forward to preserve and reuse data analysis code and frameworks as well as statistical models (see Refs. [12–14] for example). These efforts emphasize the importance of a holistic approach in preserving different kinds of digital objects. The importance of such preservation has been repeatedly iterated in literature, especially in the context of reinterpretation and reproduction of analysis results in HEP [13, 15–17].

In order to make datasets Findabile, Accessibile, Interoperable, and Reusable (FAIR), a set of data principles have been defined so that scientific datasets could be readily reused by both humans and machines [18]. Originally envisioned for preservation of scientific datasets, the FAIR principles have been interpreted in the context of different kinds of digital objects, including research software [19], notebooks [20], and machine learning models [21, 22]. Interpretation and application of FAIR principles for HEP datasets has been explored in Ref. [23]. Exploring FAIR principles in the context of DOs other than datasets is also becoming popular in HEP. For instance, the NNPDF collaboration has developed a FAIR-inspired open-source framework for parton distribution analyses [24]. Making DOs FAIR (i.e. FAIRification) requires dedicated efforts on two fronts- (i) establishing guidelines for developing new digital contents so that they conform to the FAIR principles and (ii) developing tools to allow FAIR-ification of existing digital contents. Given the diversity of digital resources used in HEP research, it is in general expected that the exact manifestation of FAIR principle-driven tools will depend on the nature of the digital object itself.

---

[1]For example, the ATLAS collaboration supports dedicated data storage facility [9] and maintenance of data processing software [10] for its members

Developing dedicated cyberinfrastructure is a novel paradigm in facilitating open and FAIR science. For instance, the `Cookiecutter` tool [25] allows structural organization of machine learning model repositories, development of containers, and publishing such codebases with persistent identifiers like digital object identifiers (DOIs). In this paper, we demonstrate the development of dedicated FAIRification tool in order to preserve `Universal FeynRules Output` (UFO) [26] models. These models are customized python libraries with a predefined format, often used to store information about beyond standard model (BSM) physics models, and can be used as plug-in libraries for MC simulation of BSM physics with event generator software. In the following section, we briefly describe the content and format of UFO models and motivate the necessity of their FAIRification. Section 3 introduces the proposed framework for making UFO models FAIR and finally, the broader outlook inspired by this work in FAIRifying UFO models is discussed in the final remarks in Scction 4.

## 2 The Need to FAIRify UFO Models

The physics program of large collider experiments like the LHC spends major efforts in search of BSM physics. These searches for BSM physics are motivated by novel theoretical extensions of the standard model (SM). The UFO model format was first introduced in Ref. [26] in order to streamline the process of matrix element calculations for BSM physics models in a generator independent way.

The UFO model format organizes the contents of the new physics as a self-contained python library. It contains a collection of *model-independent* and *model-dependent* python scripts (Table 1). While the model-indpendent scripts include general technical functionalities of the model, the model-dependent files contain physics-specific information. For instance, the script `particles.py` contains information about all fundamental particles defined within the scope of the model and their corresponding properties like color, charge, and spin quantum numbers, PDG identifier [27], parameters representing their mass and widths etc.

| Model-independent files | Model-dependent files |
|---|---|
| `__init__.py` `object_library.py` `function_library.py` `write_param_card.py` | `particles.py` `coupling_orders.py` `parameters.py` `vertices.py` `couplings.py` `lorentz.py` |

Table 1: List of model-independent and model-dependent files traditionally included in the UFO format

The primary motivation of developing the UFO format philosophically coincides with the FAIR principles. Content of BSM physics Lagrangians were generally organized as text files, whose format depended on the choice of MC generation software. Developing new physics models as python libraries enabled using the same digital format to be interoperable across different MC generation platforms like MadGraph, Herwig, and Sherpa [6, 7, 28]. Since its introduction, the UFO format has been widely used to develop phenomenological BSM physics models. At the LHC experiments, these models are regularly used to generate MC events for

BSM physics searches. In many cases, these models are developed in association with physics publications that explain and validate the underlying theoretical models. However, the preservation of these models as persistent digital contents has not been ideal. Since its inception, many of the UFO models have typically been hosted in the FeynRules Model Database [29]. Most of these models are simply hosted as compressed containers like a `.tar` or `.zip` file, without any persistent identifier, trackable version controlling, or well-documented machine readable metadata. Some models like the SMEFTsim [30] package are now maintained with modern software management tools like `git` based version controlling [31] and hosted as `Github` repositories. However, the standard FAIRification practices like use of persistent digital identifiers or developing machine-readable metadata are yet to be normative within the community. Moreover, hosting a project on `Github` or similar platforms alone does not make DOs FAIR as these repositiories are not persistent and often offer additional challenges to maintain the reliability and integrity of the DOs they host [32]. Without any well-defined guideline on community-wide agreed upon practices to preserve such digital contents, it can be difficult to validate, reproduce, reinterpret, or recast any research that relies on these models. For instance, the analysis performed by the ATLAS collaboration in Ref. [33] uses a number of UFO models for generating MC events for dark matter signal. These models were hosted by CERN's `svnweb` service[2] that used the `subversion` version controlling tool [34]. Since the discontinuation of this service and a complete shift to `gitlab` (another `git` based version controlling system) as the designated DO repository for CERN, the URLs pointed to by the corresponding references are no longer valid. Moreover, those references neither identify the authors of the model nor refer to any metadata that might allow a reader to access any information about the construction of the model. In situations like this, having no persistent identifiers associated with these contents makes it virtually impossible to identify the content or version of a UFO model used in a certain physics analysis, track changes across versions, or attribute citation credit to developers of these digital contents. Also, reusing these models to reproduce validation results or reinterpret physics analyses can become challenging. Given these UFO models directly contribute to a large number of HEP publications, ensuring reproducibility of these results require these models to be preserved in an open and FAIR way.

## 3    Cyberinfrastructure for FAIRifying UFOs

Facilitating FAIRification of UFO models must consider certain aspects of current practices of developing and using UFO models. Given that the UFO format is widely used, any tool to facilitate their FAIRification must not interfere with the processes of how to develop or use them. Also, the tool should be capable of FAIRifying pre-existing UFO models and allow distinct and persistent identification of different versions of the same model. Additionally, these tools should facilitate generation and preservation of rich metadata with proper context and details regarding the model while allow users to make use of this metadata to search for and download these models. Finally, any tool dedicately developed to FAIRify UFO models should itself be made FAIR.

Keeping these criteria in mind, the cyberinfrastructure for FAIRification of UFO models

---

[2]It used to be hosted on http://svnweb.cern.ch/. Currently, this web service is discontinued and this link is no longer active

has been developed in two independently maintained public repositories- `UFOManager` and `UFOMetadata`. The following subsections address the functionality of each of these repositories.

## 3.1  `UFOManager`

The `UFOManager` repository contains a collection of python scripts that can be used to-

- i. perform validation checks on a UFO model
- ii. develop enriched machine-readable metadata for each model
- iii. publish these models with persistent DOIs and independently preserve their metadata
- iv. upload a new version of the previously uploaded model
- v. search for models with keywords
- vi. download models according to user's choice

Among the aforementioned functionalities, (i)-(iv) are facilitated by the script `UFOUpload.py` while the remaining functionalities are supported by the script `UFODownload.py`. Many of the existing UFO models were developed using Python 2. Hence, the module `UFOUpload.py` has been developed to be compatible with both versions of Python. The `UFODownload` script, however, is only compatible with Python 3. The instructions for creating the appropriate environments and installing dependencies are included in the `README` of the package.

The `UFOUpload` and `UFODownload` scripts can be run as standalone scripts to perform the desired task. The `UFOUpload` script can be run in five independent modes, which has to be mentioned as a command line argument when running the script in standalone mode. The script can be run on the command-line interface as a simple Python script using either of the following commands-

```
python UFOUpload.py <operation-mode>
python -m UFOManager.UFOUpload <operation-mode>
```

where `<operation-mode>` refers to one of the following five strings- `Validation Check,` `Generate metadata, Upload model, Update new version,` and `Upload metadata to GitHub`. The functionality provided by each of these operation modes can be easily understood from their names. The script is designed to run interactively, so necessary inputs are requested from the user at different stages of executing the script. In order to use the `UFOUpload` script, the user must provide a list of locally accessible directories in a `.txt` file where each directory houses a single UFO model along with an initial metadata file called `metadata.json`, in the format of a `json` file. Storing metadata in the format of `json` files ensures that it is simultaneously human and machine-readable, and also compatible with multiple programming platforms ensuring its interoperability. In the process of FAIRifying these models with persistent DOIs, they are uploaded to Zenodo [35]. This also allows trackable version controlling for the models since `Zenodo` provides unique DOIs for different versions of the same entry as well as a unique *concept DOI* that can be used to locate all versions of a given content.

`UFOManager` can also be used as a dedicated Python package and can be imported and used as a Python package using the following syntax-

```
from UFOManager import UFOUpload
UFOUpload.UFOUpload(<operation-mode>,<model-list>)
```

where `<operation-mode>` represents one of the previously mentioned operation modes and `<model-list>` is the text file with a list of UFO models.

Successful publishing of model relies on creation of enriched metadata, which eventually relies on the successful completion of validation checks. Hence, any attempt to upload a new model or update the version of an existing model will automatically perform validation checks and create enriched metadata. This enriched metadata is built upon the information provided by the user in the initial `metadata.json` file. This initial metadata must contain at least three fields- (i) the names of the authors of the model along with contact information (an email address) for at least one of the authors, (ii) a brief description of the model, and (iii) a digital identifier (an arxiv ID or a DOI) for an associated publication that explains the underlying model and provides physics validation. The initial metadata can also contain additional information like the link to the webpage that hosts the model (e.g. a `Github` repository or the link to a model's location within FeynRules Model Database). The content provided in the `metadata.json` file to register one of the UFO models used to describe the physics of Vector-like Quarks (VLQs) [36] using our tool is given in Table 2.

```
{
    "Author":   [
                            {"name" :   "Luca Panizzi",
                            "affiliation":   "Uppsala University",
                            "contact":   "luca.panizzi@physics.uu.se"},
                            {"name" :   "Benjamin Fuks",
                            "affiliation":   "Sorbonne University",
                            "contact":   "fuks@lpthe.jussieu.fr"}
    ],
    "Paper_id":         {"doi":   "10.1140/epjc/s10052-017-4686-z",
                         "arXiv" :   "1610.04622"},
    "Description":      "Vector-like Quark UFO Model at NLO QCD
                         with four flavour scheme",
    "Model Homepage" :  "https://feynrules.irmp.ucl.ac.be/wiki/NLOModels"
}
```

Table 2: Content of initial `metadata.json` file used to FAIRify the UFO model in Ref. [36]

The validation check looks for the presence of required files in the UFO model, inspects the initial metadata for the necessary content and their validity (e.g. if the provided arxiv ID represents a valid entry). Then, it checks if the model can be imported as a standalone python package, since that is the intended mode of usage for these models. Afterwards, it inspects each of the independent scripts within the UFO model. These inspections include tests on whether the script can be imported as a Python module, each essential script contains a non-zero count of required objects, and some basic validation checks. These checks include validating the uniqueness and validity of the PDG IDs assigned to the particles as well as whether the model supports Next-to-Leading-Order (NLO) calculations. Validation of PDG IDs is performed using the `Particles` package [37]. During these checks, it also collects information for the enriched metadata.

Besides the fields provided with the initial metadata, the enriched metadata, which also

is stored as a `json` file, contains additional information about the number of parameters, couplings, coupling orders, vertices, lorentz tensors, propagators, and decays contained in different scripts within the model. It also contains maps of fundamental particles and their PDG IDs defined in the model. The enriched metadata also records the version of Python the model has been validated with and the version identifier of the model itself. A field for storing the model's DOI is created. If the user wishes to upload the model or update an existing model with the current content, the model is uploaded to `Zenodo` using its REST-API service. The DOI generated during uploading the model to `Zenodo` is stored in the enriched metadata. The content of the enriched metadata generated by the tool for the previously mentioned VLQ UFO model [36] is given in Table A1.

The enriched metadata is finally uploaded in the `UFOMetadata` repository. To perform these uploads the user needs to have registered accounts with both `Zenodo` and `Github` and obtain secure access tokens for the respective accounts. These access tokens allow validating the registered user accounts and use the APIs in a secure manner.

The final piece of the `UFOManager` tool is the `UFODownload` script, which allows the users to search for and access these models. It can be run as a standalone Python script from the command line interface using either of the following commands-

```
python UFODownload.py <operation-mode>
python -m UFOManager.UFODownload <operation-mode>
```

where `operation-mode` refers to one of the following strings- `Search for model`, `Download model,` and `Search and Download`. To use it within a facilitator script, it can be run as-

```
from UFOManager import UFODownload
UFODownload.UFODownload(<operation-mode>)
```

The `UFODownload` script allows the user to search for UFO models based on a number of keywords, including arxiv ID or DOI of associated publication, PDG ID of new elementary particles, DOI of the model, and model name. Table B2 shown an example of the usage of this script using the interactive command line interface to search for models with a certain PDG ID. When searching for models, it lists all available versions of the models whose metadata are available in `UFOMetadata`. It uses the `zenodo_get` package [38] to allow users to download the desired models. The script interfaces with the latest version of the `UFOMetadata` repository and offers the user a set of options based on the search criteria. Upon request, the metadata and corresponding model are downloaded by this script.

## 3.2 `UFOMetadata`

FAIRification of DOs require their metadata to include qualifying reference to the original content and be stored independently of the data [18]. Hence, the enriched metadata generated by the `UFOManager` is stored as independent `json` files in the `UFOMetadata` repository. Every time a user decides to upload their model using the `UFOManager` tool, the corresponding metadata is sent to the `UFOMetadata` repository. However, it is also possible for users to independently register their model and submit the corresponding metadata for addition to the repository. Every request to add metadata must be done via a *pull request*, which triggers an automated `Github` workflow for continuous integration. The workflow performs a validation check on the new content to ensure- (i) none of the previously existing files has been renamed,

modified, or deleted, (ii) the newly added metadata files have all the fields that are generated with the `UFOManager` tool, and (iii) newly added models have unique and valid DOIs. These pull requests are currently merged by the maintainers of repository in a regular interval to keep the repository updated with the latest additions. It should be noted that the `UFODownload` script makes use of the latest version of the `main` branch of the `UFOMetadata` repository, so access to any mew metadata is only made available after the pull requests are merged.

### 3.3   FAIRifying `UFOManager` and `UFOMetadata`

While the tools facilitated via `UFOManager` and `UFOMetadata` enable FAIRification of UFO models, it is only expected to have these tools made available as FAIR DOs themselves. The latest stable releases of these repositories are made available with persistent DOIs via `Zenodo`. The metadata can for these repositories can be accessed via URLs and obtained with JSON Schema with `Zenodo`'s REST API. The use of JSON Schema provides clear human and machine readable documentation. The following table gives the DOIs for the latest versions of these packages.

| Content | Latest Version | DOI/URL |
|---------|---------------|---------|
| `UFOManager` | 2.0.0 | 10.5281/zenodo.7066042 [39] |
| Metadata for `UFOManager` | N/A | https://zenodo.org/api/records/7066042 |
| `UFOMetadata` | 2.0.0 | 10.5281/zenodo.7066046 [40] |
| Metadata for `UFOMetadata` | N/A | https://zenodo.org/api/records/7066046 |

Table 3: DOIs for the latest stable releases of `UFOManager` and `UFOMetadata` and URLs to access their corresponding metadata

## 4   Outlook and Conclusion

With an ever-increasing volume of DOs required in mainstream HEP research, FAIR management and preservation of these contents is necessary to ensure transparent, reliable, and reproducible results. We discussed in this paper in the context of UFO models why having standardized FAIRification practices is necessary and developed a set of simple, customized tools to facilitate such practices. `UFOManager` and `UFOMetadata` rely on well-established and widely used programming tools and practices and offer a unified approach to save UFO models as FAIR DOs. Our work additionally serves as an example on developing customized, automatic FAIRification tools for DOs used in HEP research that can serve as a guide to establishing similar practices in orthodox DO management.

## Acknowledgements

We thank Benjamin Fuks for permitting the use of publicly available UFO models authored by him. We also thank Olivier Mattelaer for encouraging discussions and suggestions on improving the content of enriched metadata for UFO models. *A.R.*: This work was supported by the FAIR Data program of the U.S. Department of Energy, Office of Science, Advanced

Scientific Computing Research, under contract number DE-SC0021258. *M.S.N*: This work was supported by the National Science Foundation under OAC-1841456. *Z.W.*: This work was supported by the National Science Foundation under Cooperative Agreement OAC-1836650.

# A    Example Content of Enriched Metadata

Table A1: Content of the enriched metadata file obtained from the `Uploadv2.py` to FAIRify the UFO model in Ref. [36]

# B  Demonstration of Usage of `UFODownload`

```
$ python -m UFOManager.UFODownload 'Search for model'
Please enter you Github access token:
You can search for model with Paper_id, Model Doi, pdg code, name (of certain particles).
Please choose your keyword type:  pdg code
Please enter your needed pdg code:  6000006
Based on your search, we find models below:
Metadata file               Model Name                                                                      Paper ID    Model DOI
VLQ_v5_5FNS_NLO_UFO.V2.0.json   UFO model for Vector-like Quarks at NLO QCD with five flavor scheme (third generation)  1610.04622  10.5281/zenodo.7038823
VLQ_v5_4FNS_only3rd_NLO_UFO.json  UFO model for Vector-like Quarks at NLO QCD with four flavor scheme (third generation)  1610.04622  10.5281/zenodo.7041539
vlq_v4_4fns.json            UFO model for Vector-like Quarks at NLO QCD with four flavor scheme                1610.04622  10.5281/zenodo.6977663
VLQ_v4_5FNS_UFO.json        UFO model for Vector-like Quarks at NLO QCD with four flavor scheme                1610.04622  10.5281/zenodo.6991118
VLQ_v5_4FNS_NLO_UFO.V2.0.json  UFO model for Vector-like Quarks at NLO QCD with four flavor scheme                1610.04622  10.5281/zenodo.7038908
VLQ_v5_4FNS_NLO_UFO.V3.0.json  UFO model for Vector-like Quarks at NLO QCD with four flavor scheme                1610.04622  10.5281/zenodo.7039382
VLQ_v5_5FNS_only3rd_NLO_UFO.json UFO model for Vector-like Quarks at NLO QCD with five flavor scheme (third generation)  1610.04622  10.5281/zenodo.7041528
Do you still want to search for models?  Please type in Yes or No:  No
```

Table B2:  Sample output from an interactive command line usage of the `UFODownload` script

# C  List of UFO models FAIRified via `UFOManager` and `UFOMetadata`

The following table contains a set of example UFO models that were FAIRified by using the `UFOManager` and `UFOMetadata` tools. These models are hosted at the publicly available FeynRules Model Database [29].

| Model Name | Version | DOI | Metadata |
|---|---|---|---|
| universal framework for t-channel dark matter models [42] | 1.0 | 10.5281/zenodo.6991108 [41] | DMsimpt_NLO_v1_2_UFO |
| Spin-2 simplified model [43] | 1.0 | 10.5281/zenodo.6908818 [44] | DMspin2 |
| NLO predictions in supersymmetric QCD [45] | 1.0 | 10.5281/zenodo.6991116 [46] | SUSYQCD_UFO |
| Vector-like Quarks at NLO QCD with five flavor scheme [48] | 1.0 | 10.5281/zenodo.6991118 [47] | VLQ_v4_5FNS_UFO |
|  | 2.0 | 10.5281/zenodo.7038823 [49] | VLQ_v5_5FNS_NLO_UFO.V2.0 |
| Vector-like Quarks at NLO QCD with five flavour scheme (third generation) [48] | 1.0 | 10.5281/zenodo.7041528 [50] | VLQ_v5_5FNS_only3rd_NLO_UFO |
| Vector-like Quarks at NLO QCD with four flavor scheme [48] | 1.0 | 10.5281/zenodo.6977663 [51] | vlq_v4_4fns |
|  | 2.0 | 10.5281/zenodo.7038908 [52] | VLQ_v5_4FNS_NLO_UFO.V2.0 |
| Vector-like Quarks at NLO QCD with four flavour scheme (third generation) [48] | 1.0 | 10.5281/zenodo.7041539 [53] | VLQ_v5_4FNS_only3rd_NLO_UFO |
| NLO predictions for sgluon pair production [54] | 1.0 | 10.5281/zenodo.6991112 [55] | sgluons_NLO |
| stop pair production to ttbar and missing energy [54] | 1.0 | 10.5281/zenodo.6991114 [56] | stop_ttmet_NLO |
| pseudo-Nambu-Goldstone Dark Matter [57] | 1.0 | 10.5281/zenodo.7065996 [58] | SM_with_pNG_UFO |
|  | 2.0 | 10.5281/zenodo.7066011 [59] | SM_with_pNG_UFO_py3.V2.0 |

Table C3:  DOIs for some of the UFO models made FAIR by the `UFOManager` and `UFOMetadata` tools

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
