# Peer review of "Making Digital Objects FAIR in High Energy Physics: An Implementation for Universal FeynRules Output (UFO) Models"

_SciPost Physics Codebases, doi:SciPost Phys. Codebases 13-r2.0 (2023) , SciPost Phys. Codebases 13 (2023)_

## Round 2 · Referee Report · Anonymous (Referee 1) · 2022-12-5

Strengths

Contributes to the FAIR-ification of digital material in HEP

Weaknesses

Currently of very limited use, not integrated into any database, implementation merits improvement, in particular the "F" in FAIR hasn't been demonstrated. Overview and references given in the introduction aren't very well researched.

Report

The present paper addresses the treatment of domain-specific digital content in particle physics in order to render it "FAIR". The concrete example considered are UFO models, for which python scripts for generating and searching metadata are presented. This is interesting and potentially useful code development, but does not meet the acceptance requirements for SciPost Physics. After appropriate revision, the paper might be suitable for SciPost Physics Codebase, though. Aspects to improve include:

  • The whole setup is very simple, consisting of standalone python scripts for upload and download . No functionality as a python library is foreseen, and for different python versions one has to call different scripts. The script names don't even convey that one is treating UFO models. All this is limiting the usability within a larger framework.

  • Requirements and dependencies aren't listed in the paper. Even the readme on github only states "Necessary Python packages need to be installed".

  • It is not clear why a virtual environment is demanded. In my opinion it should not be necessary.

  • The validation checks are too simple. Checking whether each particle is assigned a unique ID isn't sufficient. One should check whether the codes used follow the official PDG numbering scheme and whenever this is not the case, a warning should be issued. Moreover, the fundamental particles in the model should be split into known (SM) particles, new (BSM) particles with standard PDG naming, and new particles with new PDG-like codes.

  • The naming and short model descriptions in Table B2 lack standardisation. ("UFO model for..." isn't necessary in the name; one dark matter simplified model is identified as such, the other isn't; "vector-like quark" is sometimes written out and sometimes abbreviated as VLQ; similar for the flavor scheme; versions 1.0 and 2.0 of the same model have totally different Metadata names; etc). Also, why are there so few models "FAIRified"?

  • In the introduction, the description of HEPData and "dedicated data analysis frameworks" is inaccurate and needs revision; the list of refs [11-13] is incomplete; refs [14,15] should also include [12] and the recent Snowmass white paper on data and analysis preservation.

  • Section 2: The UFO format is intended as a standard interface. There are a number of MC tools beyond MadGraph which can use the UFO format. The statement that "UFO models have been demonstrated to be compatible with other MC generators as well" is misleading.

  • Still section 2: "generate the simulated MC events" ... are there non-simulated MC events?

Requested changes

See report. In particular

  • provide the UFOManager as a python tool where upload and download functionalities as well as python2 or python3 can be given as arguments;

  • clearly describe requirements and dependencies, and make the UFOManager importable as a library;

  • improve validation checks;

  • extend and standardise the list of models in Table B2;

  • include a demonstration of how to query for models with specific properties and/or particle content;

  • revise sections 1 and 2 as indicated in the report;

  • there are a number of typos to be corrected.

  • validity: low
  • significance: low
  • originality: low
  • clarity: ok
  • formatting: reasonable
  • grammar: good

Author:  Avik Roy  on 2023-02-03  [id 3307]

(in reply to Report 1 on 2022-12-05)
Category:
answer to question

We would like to thank the editor for arranging the review of this paper and the reviewer for their insightful comments and suggestions. Based on the comments we have received during review, this revised version is submitted to SciPost Physics Codebases. The following is a detailed list of changes accommodated in the current submission based on the responses we have received-

provide the UFOManager as a python tool where upload and download functionalities as well as python2 or python3 can be given as arguments; clearly describe requirements and dependencies, and make the UFOManager importable as a library;

Response: We have implemented UFOManager as a python package (while retaining the simplistic, standalone use of Upload and Download scripts). It internally determines the Python version being used and uses the scripts accordingly. Specific dependencies provided as requirements_Python2.txt and requirements_Python3.txt can be used to build dedicated conda environments (or virtual environment) based on the needs of the user. Details about some of the specific dependencies are detailed in a separate section of the current README

improve validation checks

Response: Following recommendations above, validation checks now include checking the validity of PDG code assigned to each particle as well as checking that the particle’s spin and charge conform to the expected spin and charge of the particle based on the assigned PDG code. Besides the dictionary of “All particles”, Metadata now contains dictionaries of “SM particles”, “BSM particles with valid PDG codes” and “BSM particles with non-standard PDG codes”. It also includes an additional flag to determine whether a model allows NLO calculations.

extend and standardise the list of models in Table B2;

Response: We have standardized the model namings in table C3 (which is Table B2 from the previous version). The model names also represent the title of the corresponding digital entry in Zenodo. Hence, we keep the prefix phrase “UFO Model for” with these titles to ensure that the content-type of these entries is clear to any user who comes across them. To avoid redundancy in the paper, we have omitted this prefix from the table entries in Table C3. Our implementation serves as a demonstrative example of FAIRifying UFO models. Since we don’t own the individual models, we cannot arbitrarily FAIRify models authored by others. Table B2 only includes FAIRified UFO models for which we have received permission to use our tools to demonstrate the FAIRification of models.

include a demonstration of how to query for models with specific properties and/or particle content;

Response We have added a demonstration of searching for models using PDG ID in Table B2.

revise sections 1 and 2 as indicated in the report;

**Response: ** We have added the requested revisions detailed in the report. These include-

Modified description of HEPData: We have rephrased the description of HEPData as “For instance, the Durham High-Energy Physics Database (HEPData)~\cite{hepdata} is an open-access repository established for preserving and sharing scattering data from HEP experiment, containing digitized details of plots and tables from thousands of physics analysis publications.” This description closely follows the “About HEPData” section in the HEPData website. In the following line “dedicated data analysis frameworks” has been replaced by “data analysis code and frameworks” to reflect that it refers to RECAST implementation of analysis codes and frameworks developed for individual analyses.

Added/Modified references: Ref 13 (previously 12) has been added to the reference list of [15,16] (previously [14,15]), also the Snowmass report has been cited (Ref. 17).

Removed confusing statement about UFO compatibility: We have removed the sentence "UFO models have been demonstrated to be compatible with other MC generators as well" to avoid any confusion. The immediately preceding line has been rephrased as "Developing new physics models as python libraries enabled using the same digital format to be interoperable across different MC generation platforms like MadGraph, Herwig, and Sherpa [6,7,28]."

*Other: * Changed “simulated MC events” to “MC events” in section 2

there are a number of typos to be corrected.

Response: The entire manuscript has been carefully checked once more for typos and grammatical mistakes and those issues have been fixed.

---

## Round 3 · Referee Report · Anonymous (Referee 1) · 2023-3-15

Report
The authors have addressed most of the points of the first referee report to satisfaction. A few minor points remain to be corrected in the manuscript as listed below. Once this is taken care of, I recommend publication in SciPost Physics Codebases.
-
Clickable links are provided throughout the text for some repositories or tools, but not for all. For example, links are given for NNPDF Collaboration, github and Zenodo, but not for HEPData or MC simulation tools. This should be systematically.
-
The introduction states that "While collaborations facilitate the internal preservation of their data and common software frameworks for collaboration-wide usage, preservation of digital resources independently de- veloped by smaller research groups or individuals is equally important to be able to reproduce the results from HEP research. " This gives the impression that data and software preservation by large experimental collaborations is for collaboration-internal use only. This is utterly misleading to my mind. There is huge benefit of making this material open to the public; results from HEP research shouldn't be reproducible only within the collaborations but by the whole HEP community. Please revise!
-
Refs. [12–14] are still incomplete, but since the text says "for example" it can be acceptable.
-
The acronym BSM is defined more than once.
Anonymous on 2023-02-03 [id 3306]
We would like to thank the editor for arranging the review of this paper and the reviewer for their insightful comments and suggestions. Based on the comments we have received during review, this revised version is submitted to SciPost Physics Codebases. The following is a detailed list of changes accommodated in the current submission based on the responses we have received-
Response: We have implemented UFOManager as a python package (while retaining the simplistic, standalone use of Upload and Download scripts). It internally determines the Python version being used and uses the scripts accordingly. Specific dependencies provided as
requirements_Python2.txt
andrequirements_Python3.txt
can be used to build dedicated conda environments (or virtual environment) based on the needs of the user. Details about some of the specific dependencies are detailed in a separate section of the current READMEResponse: Following recommendations above, validation checks now include checking the validity of PDG code assigned to each particle as well as checking that the particle’s spin and charge conform to the expected spin and charge of the particle based on the assigned PDG code. Besides the dictionary of “All particles”, Metadata now contains dictionaries of “SM particles”, “BSM particles with valid PDG codes” and “BSM particles with non-standard PDG codes”. It also includes an additional flag to determine whether a model allows NLO calculations.
Response: We have standardized the model namings in table C3 (which is Table B2 from the previous version). The model names also represent the title of the corresponding digital entry in Zenodo. Hence, we keep the prefix phrase “UFO Model for” with these titles to ensure that the content-type of these entries is clear to any user who comes across them. To avoid redundancy in the paper, we have omitted this prefix from the table entries in Table C3. Our implementation serves as a demonstrative example of FAIRifying UFO models. Since we don’t own the individual models, we cannot arbitrarily FAIRify models authored by others. Table B2 only includes FAIRified UFO models for which we have received permission to use our tools to demonstrate the FAIRification of models.
Response We have added a demonstration of searching for models using PDG ID in Table B2.
**Response: ** We have added the requested revisions detailed in the report. These include-
Modified description of HEPData: We have rephrased the description of HEPData as “For instance, the Durham High-Energy Physics Database (HEPData)~\cite{hepdata} is an open-access repository established for preserving and sharing scattering data from HEP experiment, containing digitized details of plots and tables from thousands of physics analysis publications.” This description closely follows the “About HEPData” section in the HEPData website. In the following line “dedicated data analysis frameworks” has been replaced by “data analysis code and frameworks” to reflect that it refers to RECAST implementation of analysis codes and frameworks developed for individual analyses.
Added/Modified references: Ref 13 (previously 12) has been added to the reference list of [15,16] (previously [14,15]), also the Snowmass report has been cited (Ref. 17).
Removed confusing statement about UFO compatibility: We have removed the sentence "UFO models have been demonstrated to be compatible with other MC generators as well" to avoid any confusion. The immediately preceding line has been rephrased as "Developing new physics models as python libraries enabled using the same digital format to be interoperable across different MC generation platforms like MadGraph, Herwig, and Sherpa [6,7,28]."
*Other: * Changed “simulated MC events” to “MC events” in section 2
Response: The entire manuscript has been carefully checked once more for typos and grammatical mistakes and those issues have been fixed.

---

## Round 3 · List of Changes

- The code repository UFOManager has been modified to be usable as a Python package, the corresponding discussion in the manuscript about how to use this package has been accordingly modified
- Instructions to setup the necessary environments to run this package have been updated
- Validation checks have been improved. Additional checks to verify the validity of PDG IDs have been added. The enriched metadata also contains additional dictionaries to separately accommodate SM particle, BSM particles with valid PDG IDs, and BSM particles with PDG-like IDs
- A demonstration of using UFOManager to search for models has been added in Table B2
- List of FAIRified UFO models in table C3 (previously B2) has been standardized
- Some of the text and references in section 1 and 2 have been modified following the referee's suggestions
- Some typos and grammatical errors have been fixed

---

## Round 4 · List of Changes

- Included clickable link for HEPData, ROOT, MadGraph, Sherpa, Pythia, and Herwig
- Modified the statement about collaboration-maintained data and software to "While collaborations facilitate the preservation of their data and common software frameworks for usage both within and outside of the collaboration1, preservation of digital resources independently developed by smaller research groups or individuals is equally important to be able to reproduce the results from HEP research."
- Removed the second instance of introducing the acronym BSM

---

## Editorial Decision

published